# Choice and impact of EQ-5D-5L value set in cost-utility analyses alongside multinational trials: Insights from PREFERABLE-EFFECT and CONVINCE

Aniek E. M. Schouten[1]*, Geert W. J. Frederix[1,2], Martijn M. Stuiver[3], Felix Fischer[4], Anouk E. Hiensch[1], Krister Cromm[5,6], Bernard Canaud[5,7], Giovanni F. M. Strippoli[8,9], Anne M. May[1,3], Miriam P. van der Meulen[1], On behalf of the PREFERABLE-EFFECT Scientific Committee, and the CONVINCE Scientific Committee[¶]

1 Department of Epidemiology and Health Economics, Julius Center for Health Sciences and Primary Care, University Medical Center Utrecht, Utrecht University, Utrecht, the Netherlands, 2 Digital Transformation in Rehabilitation Care, HAN University of Applied Sciences, Nijmegen, the Netherlands, 3 Division of Psychosocial Research and Epidemiology & Center for Quality of Life, Netherlands Cancer Institute, Amsterdam, the Netherlands, 4 Center for Patient-Centered Outcomes Research, Department for Psychosomatic Medicine, Charité, Universitätsmedizin Berlin, corporate member of Freie Universität Berlin and Humboldt Universität zu Berlin, Berlin, Germany, 5 Fresenius Medical Care Deutschland GmbH, Global Medical Office, Bad Homburg, Germany, 6 Department of Psychosomatic Medicine, Charité – Universitätsmedizin Berlin, corporate member of Freie Universität Berlin and Humboldt-Universität zu Berlin, Berlin, Germany, 7 Montpellier University School of Medicine, Montpellier, France, 8 Department of Precision and Regenerative Medicine and Ionian Area, University of Bari, Bari, Italy, 9 School of Public Health, University of Sydney, Sydney, Australia

¶ The complete membership of the author group can be found in the Acknowledgments
* a.e.m.schouten-10@umcutrecht.nl

## Abstract

### Objectives

Multinational clinical trials are increasingly common in the European Economic Area, yet no guideline exists on which EQ-5D value set(s) should be used for health economic evaluations alongside multinational trials. Either a single value set or country-specific value sets can be applied, and previous studies have shown that the choice of value set can impact the estimated utility scores. For the EQ-5D-5L, the impact on European pooled cost-utility analyses has not been established. This study evaluates the impact of EQ-5D-5L value sets on cost-utility outcomes in two multinational trials.

### Methods

Data from two multinational randomized controlled trials were used: (i) supervised exercise compared to usual care for patients with metastatic breast cancer (PREFERABLE-EFFECT), (ii) hemodiafiltration compared to hemodialysis for patients with kidney failure (CONVINCE). EQ-5D-5L was assessed at baseline and during follow-up. Utility scores and quality-adjusted life years (QALYs) were

**Data availability statement:** All data requests can be sent to the following institutional email address: HTAsecretariaat@umcutrecht.nl. Requests for the R scripts of this methodological article can be directed to the corresponding author. The data that support the findings of this study are not openly available owing to reasons of confidentiality. Researchers can direct requests for the PREFERABLE-EFFECT data to Professor Anne May. Requests should also be sent to a.m.may@umcutrecht.nl. Pseudonymized data (including data dictionaries) will be made available through the Digital Research Environment, which is a trusted digital research environment that can be accessed at https://mydre.org. This will be carried out after the review and approval of a methodologically sound proposal by the General Assembly of PREFERABLE, with a signed data access agreement, which is in line with Ethics Committee requirements (The Ethics Committee of University Medical Center Utrecht, The Netherlands). Requests will be processed within 6 weeks. These files will be available from the date of publication until the date stated in the approved request. Once the PREFERABLE project has been fully completed, the database will be anonymized and shared using DataverseNL. The study protocol is available as an open access publication (https://doi.org/10.1186/s13063-022-06556-7). Individual investigators can request CONVINCE data by sending a proposal directed to the Convince Scientific Committee. Proposals should also be sent to r.w.m.vernooij-2@umcutrecht.nl. To gain access to data, requestors will need to sign a data access agreement. Proposed use of data can be made available for the complete deidentified patient data set, along with study protocol, statistical analyses plan, informed consent, analytical code (R scripts), and Excel document with cohort model.

**Funding:** The author(s) received no specific funding for this work. PREFERABLE-EFFECT was supported by the European Union's Horizon 2020 research and innovation program (grant agreement 825677) and the National Health and Medical Research Council of Australia (2018/GNT1170698). CONVINCE was exclusively supported by a grant from the

calculated with the country-specific value set for each participant, and with the value set of each included country. The probability of cost-effectiveness was estimated using bootstrapping.

### Results

Mean assigned utility scores per year alive ranged between 0.762 and 0.889 for PREFERABLE-EFFECT, and 0.673 and 0.806 for CONVINCE. The difference in utility scores is largest when participants report low quality of life. Estimated QALY gains ranged between 0.013 and 0.020 for PREFERABLE-EFFECT and 0.045 and 0.058 for CONVINCE. The maximum difference in probability of cost-effectiveness between the value sets was Δ8.3% at €80,000/QALY in PREFERABLE-EFFECT, and Δ11.1% at €40,000/QALY for CONVINCE.

### Conclusions

Choice of value set led to substantial variation in absolute utility scores and QALYs, which may influence cost-utility outcomes. This impact could be greater when an intervention prevents or aids recovery of health conditions associated with low quality of life, or results in large mortality differences. Scenario analyses using multiple value sets should be conducted for multinational trials.

---

## Introduction

The EQ-5D is an instrument for estimating health-related quality of life (HRQoL) [1]. This instrument is frequently used to calculate the effect outcome 'quality-adjusted life years' (QALYs) in health economic evaluations [2,3], and is preferred in the majority of national guidelines [4,5]. People's perceived problems with mobility, self-care, usual activities, pain/discomfort, and anxiety/depression are estimated using a questionnaire with either a 3-level or a 5-level scale (EQ-5D-3L and EQ-5D-5L, respectively). These descriptive answers can be translated into a utility score using a value set [6]. This utility score reflects how good or bad someone perceives their health status to be, with "1" being perfect health and "0" death [1], with negative values indicating a health state considered worse than death. By combining utility scores with length of life, QALYs can be estimated, whereby one QALY equals one year of perfect health. Use of EQ-5D-5L is preferred over the EQ-5D-3L, due to improved psychometric performance [7], and more recently derived value sets using a standardized protocol [8,9].

Country-specific value sets have been developed for both the EQ-5D-3L and the EQ-5D-5L to capture local preferences of different health states [10]. Health economic evaluation guidelines recommend using the value set of the jurisdiction of interest for single-country studies [11]. However, there is currently no guideline on which value set(s) should be used for (pooled) multinational health economic evaluations. Over recent years, there has been an increase in multinational clinical trials

European Commission (Horizon 2020 program, grant agreement 754803).

Competing interests: GF reports a consulting role for Illumina (Inst). KC reports holding Fresenius Medical Care company stocks (ordinary shares) and being an employee of Fresenius Medical Care. BC reports being CEO of MTX Consulting Int, Montpellier; Emeritus Medical Officer at FMC Deutschland Global Medical Office; and Emeritus Professor of Medicine, Montpellier, France. GS reports receiving honoraria for educational events from Fresenius Medical Care Deutschland GmbH. All other authors declared no competing interests. This does not alter our adherence to PLOS ONE policies on sharing data and materials.

relative to single-country trials, particularly in the European Economic Area [12]. Statistical power in multinational trials is often not sufficient to perform country-specific cost-utility subgroup analyses [11,13,14]. For pooled analyses, there is no standard practice for estimating utilities [15]. In practice, a single value set is most commonly applied, often that of the UK as it was the first developed value set [6,16]. An alternative is applying the country-specific value set of each participant. A single value set, reflecting the preferences of a larger region, has been proposed as solution [17–20], but this is currently not commonly adopted [10].

Previous studies have shown that choice of value set can impact the absolute value of estimated utility scores [16,21–31]. Moreover, van Dongen et al. (2021) showed that both incremental effects and probability of being cost-effective could vary significantly between value sets [23]. These differences were particularly large between countries of different continents (e.g., between Zimbabwe and Taiwan, or Canada and Singapore). Oppong et al. (2010) found that the choice between European, country-specific, or UK value sets in country-specific sub-group analyses had little impact on cost-utility outcomes when using the EQ-5D-3L [16]. For the EQ-5D-5L, the impact on European pooled multinational cost-utility analyses has not yet been established.

This study aims to evaluate the impact of the choice of EQ-5D-5L value set on pooled cost-utility outcomes alongside European multinational trials, using data from two studies involving patients with chronic diseases. The two studies include distinct patient populations (patients with metastatic breast cancer and kidney failure), to assess whether the observed impact is consistent across different demographic groups. Utility scores per year alive and QALYs, estimated using both single and country-specific value sets, were assessed for the entire study population, as well as stratified by country and randomization group. The findings provide evidence to support informed decision-making for health economic evaluations conducted alongside multinational clinical trials.

## Methods

The impact of EQ-5D-5L value sets on cost-utility outcomes was estimated using collected EQ-5D-5L data from two multinational, randomized controlled trials: PREFERABLE-EFFECT and CONVINCE. Cost-effectiveness analyses of these trials have been performed [32,33]. Utility scores of the full sample were calculated with the value sets of each included country, and with the country-specific value set of each participant. Differences in utility scores per year alive and quality-adjusted life years (QALYs) were evaluated at study population level and randomization level. The probability of cost-effectiveness was estimated using bootstrapping.

### Study population

The PREFERABLE-EFFECT trial (ClinicalTrials.gov, NCT04120298) included patients with metastatic breast cancer (diagnosis of breast cancer stage IV) from 8 sites in 6 different countries. Participating countries were the Netherlands (NL), Germany (DE), Spain (ES), Sweden (SE), Poland (PL) and Australia (AU) (S1 Table

in S1 File). The intervention group received a structured and individualized exercise program in addition to usual care for 9 months, while the control group received usual care, as well as general physical activity advice and an activity tracker. Recruitment took place from January 8th, 2020 to August 3rd, 2022. Participants (N = 357) were randomized to either intervention (N = 178) or control (N = 179) and data was collected for 9 months. Patients valued their health status with the EQ-5D-5L at baseline and in 3-month increments. Majority of patients were female (99%) and average age was 55.4 (SD = 11.1) years old. The trial protocol, primary study results and cost-effectiveness analysis have been published elsewhere [33–35].

The CONVINCE trial (Netherlands National Trial Register, NTR 7138) recruited patients with kidney failure who had received hemodialysis (HD) for at least 3 months. Patients were included in 61 centers in 8 different countries. Participating countries were the Netherlands (NL), Germany (DE), Spain (ES), France (FR), the United Kingdom (UK), Portugal (PT), Hungary (HU) and Romania (RO) (S1 Table in S1 File). The intervention group received high-dose hemodiafiltration (HDF), while the control group received high-flux hemodialysis. Recruitment took place from November 7th, 2018, until March 24th, 2021. Participants (N = 1360) were randomized to either intervention (N = 677) or control (N = 683) and data of participants who completed the trial were collected for at least 24 months. Patients valued their health status with the EQ-5D-5L at baseline and in 6-month increments. Majority of patients were male (62.9%) and average age was 62.4 (SD = 13.5) years old. The trial protocol, primary study results and cost-effectiveness analysis have been published elsewhere [32,36,37].

The study protocols of both PREFERABLE-EFFECT (19–524/M) and CONVINCE (18–260/D) were approved by the institutional review board of the University Medical Center Utrecht, the Netherlands, and by the local ethical review boards of all participating institutions. The studies were conducted in accordance with Good Clinical Practice and the Declaration of Helsinki. All patients of both PREFERABLE-EFFECT and CONVINCE provided written informed consent.

## Data analysis

Data analysis was performed with R (R Foundation for Statistical Computing), version 4.4.1. Across all timepoints, 10.7% and 23.9% of the EQ-5D-5L data of PREFERABLE-EFFECT and CONVINCE were missing in total, respectively. Missing values were predicted using multiple imputation (m = 20) [38], applying the R mice package (see S2 Table in S1 File for predictors) [39]. The imputed datasets were analyzed using Rubin's rules, a method that combines (pools) results from multiple imputed datasets and adjusts the standard error to reflect the uncertainty introduced by the missing data [40].

Unlike the published cost-effectiveness analyses of both trials, the EQ-5D domains were imputed rather than the calculated utility scores to ensure we investigated the difference in value sets, not the difference in imputation. Based on the imputed domains, utilities were estimated multiple times for the entire study population by applying the EQ-5D-5L value set of each included country of the study separately [41–51], as well as the appropriate country-specific value set per patient (CS). For the UK, the crosswalk version of the EQ-5D-3L value set was applied using the index values at the EuroQol website [52,53], following the UK national guideline [54]. A new value set for the EQ-5D-5L has been published in March 2026, which we have included as additional analysis [55]. While it has been proposed that this new EQ-5D-5L value set will replace the EQ-5D-3L cross-walk version as the preferred value set in the UK, this is not yet the case at the time of writing [56]. Therefore, results for both value sets are presented. The utility score was set to 0 for patients who died from date of death onwards. QALYs were calculated using an area under the curve approach with linear interpolation [57].

Descriptive statistics of baseline utility scores, utility per year alive and QALYs were calculated, reporting the mean values of the entire study population, per country, and per randomization group. Uncertainty was estimated by bootstrapping (5000 iterations for each of the 20 imputed data sets) with two seemingly unrelated regressions (SUR) to correct for confounders [58]. For each value set, the probability of cost-effectiveness was explored using the lowest and highest willingness-to-pay thresholds of the Netherlands (i.e., €20,000/QALY to €80,000/QALY) [59], as well as the thresholds in between. For CONVINCE, results were also extrapolated over lifetime using a Markov cohort model with a cycle length

of 1 year, simulating 1000 patients. The model contained three health states: dialysis (HD or HDF), renal transplant, and dead. Half-cycle correction was applied. This model is the same as in the cost-effectiveness article where more details can be found [32], apart from the utility scores.

### Scenario analyses

A scenario analysis was conducted to facilitate interpretation of the magnitude of impact related to choice of value set. Additional multiple imputation approaches were evaluated, all of which could be used in practice. First, only the seed was changed, with three different seed variations [60]. Second, the EQ-5D-5L answers were translated into utility scores first and then imputed, also with three different seed variations.

For all scenarios, the difference in utility alive between intervention and control per value set was calculated (i.e., utility alive gain). We then assessed: (1) the difference between the maximum and minimum utility alive gain across the different value sets *for each imputation scenario*; (2) the difference between the maximum and minimum utility alive gain across the different imputation scenario *for each value set*.

## Results

### EQ-5D-5L domain outcomes over time

For both studies, quality of life decreased across all EQ-5D-5L domains between baseline and last follow-up (S3 Table in S1 File). The domain most impacted at baseline was pain/discomfort, with 66.6% and 62.6% of participants of PREFERABLE-EFFECT and CONVINCE, respectively, indicating experiencing moderate, severe or extreme problems. Imputing the missing values at 9 months resulted in a smaller proportion of 'no problem' scores across both studies and all domains, suggesting that health-related quality of life is likely lower for participants that do not fill in the questionnaires.

### Utilities across value sets, entire study population

First, utilities were estimated with the different value sets for the entire study population of both trials, without stratification. Table 1 shows the differences between the highest and lowest baseline utility scores ($\Delta_{max}$0.121 and $\Delta_{max}$0.133, respectively) and utility score per year alive ($\Delta_{max}$0.127 and $\Delta_{max}$0.133, respectively) when applying the different value sets for PREFERABLE-EFFECT and CONVINCE. In PREFERABLE-EFFECT, the mean QALYs ranged between 0.555 QALY (NL value set) and 0.647 QALY (PL value set). In CONVINCE, mean QALYs ranged between 1.287 QALY (UK 3L value set) and 1.537 QALY (RO value set). Using the country-specific value sets of each participant resulted in utility scores close to the average.

Fig 1 shows the distribution of the baseline utility scores of both study populations using the different value sets in violin plots (see S1 & S2 Figs in S1 File for the distributions of QALYs and utility scores per year alive). The plots are skewed to higher utility scores for all value sets, but most for the Swedish and Polish value sets in PREFERABLE-EFFECT, and the French value set in CONVINCE. Fig 2 shows the difference between the German value set (close to the average utility score across value sets) and other value sets from low to high utility scores. Most variation between value sets occurs at the lower utility scores, while variation is smaller for the higher utility scores, indicating that the impact of choice of value set is larger at lower quality of life scores.

### Utilities across value sets, participants stratified by country

Mean utility alive scores of patients split up by country are shown in Table 2, and S4 Table in S1 File. For PREFERABLE-EFFECT, the Swedish and Polish patients have the lowest mean utility alive scores when using the German value set, while the Dutch and the Spanish patients have the highest mean utility alive scores. For CONVINCE, patients from the UK and Germany had the lowest utility alive scores when using the German value set, and patients from Hungary and

**Table 1. Mean baseline utility scores, quality adjusted life years and utility alive scores of complete patient population using different value sets.**

| | Baseline utility score [1] | Quality-adjusted life years [2] (QALYs) | Utility alive score [2] |
|---|---|---|---|
| **PREFERABLE EFFECT** | | | |
| DE value set | 0.880 | 0.608 | 0.834 |
| NL value set | 0.808 | 0.555 | 0.762 |
| ES value set | 0.832 | 0.575 | 0.789 |
| SE value set | 0.920 | 0.638 | 0.876 |
| PL value set | 0.929 | 0.647 | 0.889 |
| AU value set | 0.898 | 0.620 | 0.851 |
| CS value set | 0.873 | 0.604 | 0.829 |
| **Difference highest-lowest** | **0.121 utility** | **0.092 QALY** | **0.127 utility** |
| **CONVINCE** | | | |
| DE value set | 0.813 | 1.471 | 0.771 |
| NL value set | 0.746 | 1.348 | 0.705 |
| ES value set | 0.767 | 1.389 | 0.727 |
| PT value set | 0.804 | 1.453 | 0.761 |
| FR value set | 0.849 | 1.535 | 0.806 |
| UK 3L value set[3]<br>UK 5L value set[3] | 0.716<br>0.796 | 1.287<br>1.437 | 0.673<br>0.753 |
| HU value set | 0.783 | 1.408 | 0.736 |
| RO value set | 0.845 | 1.537 | 0.807 |
| CS value set | 0.805 | 1.459 | 0.765 |
| **Difference highest-lowest** | **0.133 utility** | **0.250 QALY** | **0.133 utility** |

DE, Germany; NL, the Netherlands; ES, Spain; SE, Sweden; PL, Poland; AU, Australia; CS, country-specific; PT, Portugal; FR, France; UK, United Kingdom; HU, Hungary; RO, Romania.

[1]For visualization of uncertainty, see violin plots with box plots in Fig 1. These provide a more accurate representation than standard deviations or standard errors due to the skewness of utility distributions.

[2]For visualization of uncertainty, see violin plots with box plots in S1 & S2 Figs in S1 File. These provide a more accurate representation than standard deviations or standard errors due to the skewness of utility distributions.

[3]For the UK, the crosswalk version of the EQ-5D-3L value set was applied using the index values at the EuroQol website, following the national guideline. As a sensitivity analysis, we also applied the EQ-5D-5L UK value set, although this is not currently endorsed for national use. For the country-specific value set, the EQ-5D-3L crosswalk value is used.

Portugal the highest. With the country-specific value set the order of the countries in terms of utility score (highest>lowest) changes. For PREFERABLE-EFFECT, the ranking changed from (ES>NL>DE>AU>PL>SE) with the German value set to (PL>AU>DE>SE>ES>NL) with the country-specific value set. For CONVINCE, the ranking changed from (HU>PT>RO>ES>FR>NL>DE>UK) with the German value set to (RO>FR>HU>PT>DE>ES>NL>UK) with the country-specific value set.

## Utilities across value sets, participants stratified by randomization group

After bootstrapping and correcting for control variables, the mean QALYs are higher for the intervention group than the control group in both studies, regardless of value set (Table 3). This is both due to more life years; 0.725 for intervention and 0.710 for control in PREFERABLE (max 0.75 year), and 1.870 for intervention and 1.845 for control in CONVINCE (max 2 years), as well as the higher utility scores alive for the intervention group (S5 Table in S1 File).

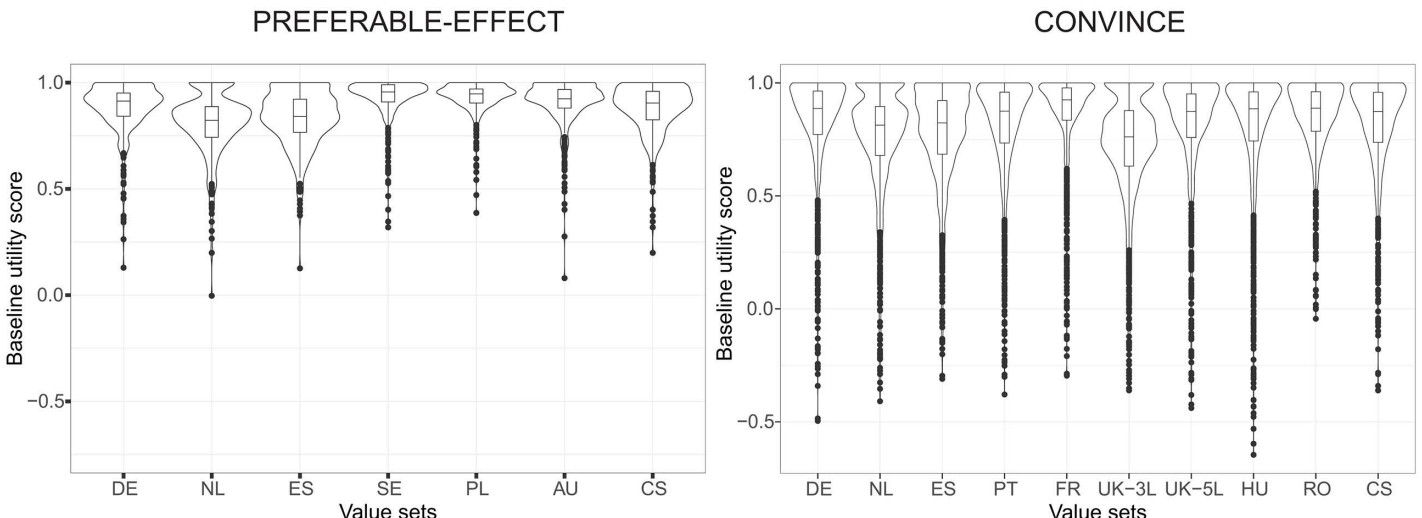

**Fig 1. Violin plots with box plots of baseline utility scores of complete patient population using different value sets.** For the UK, the crosswalk version of the EQ-5D-3L value set was applied using the index values at the EuroQol website, following the national guideline. As an additional analysis, we also applied the 2026 EQ-5D-5L UK value set, although this is not officially adopted for national use at time of writing. For the country-specific value set, the EQ-5D-3L crosswalk value is used. DE, Germany; NL, the Netherlands; ES, Spain; SE, Sweden; PL, Poland; AU, Australia; CS, country-specific; PT, Portugal; FR, France; UK[1], United Kingdom; HU, Hungary; RO, Romania.

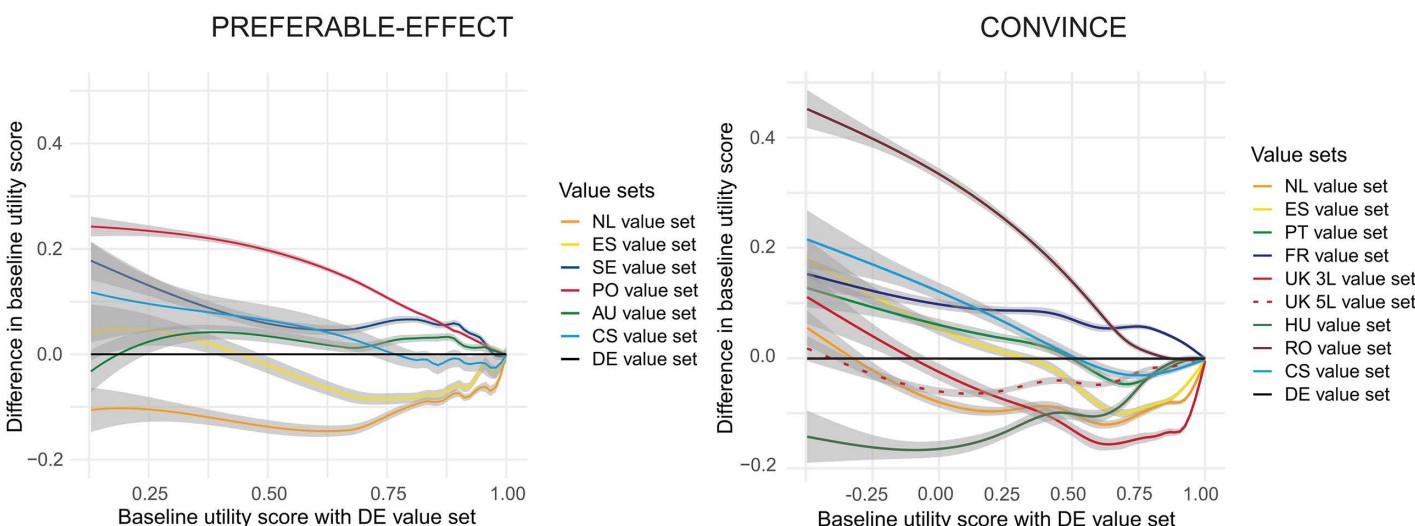

**Fig 2. Difference in baseline utility score for the entire patient population of different value sets compared to the utility score when using German value set.** Fig 2 represents the differences in baseline EQ-5D-5L utility scores between the various value sets and the German value set (Y-axis) across the range of baseline scores estimated using the German value set (X-axis). On the Y-axis, positive values indicate higher utility scores compared to the DE value set, negative values indicate lower scores. Estimates are pooled per participant aross the imputation sets. Lines are estimated using LOESS (Locally Estimated Scatterplot Smoothing), and represent a smoothed trend line with a 95% confidence interval (CI). DE, Germany; NL, the Netherlands; ES, Spain; SE, Sweden; PL, Poland; AU, Australia; CS, country-specific; PT, Portugal; FR, France; UK, United Kingdom; HU, Hungary; RO, Romania.

**Table 2. Mean utility alive scores with the country-specific value set and the German value set.**

| Mean utility alive | DE patients | NL patients | ES patients | SE patients | PL patients | AU patients | | |
|---|---|---|---|---|---|---|---|---|
| *PREFERABLE* | | | | | | | | |
| N | 89 | 91 | 54 | 44 | 44 | 35 | | |
| DE value set | 0.812 | 0.856 | 0.881 | 0.783 | 0.810 | 0.851 | | |
| CS value set | 0.812 | 0.791 | 0.844 | 0.834 | 0.885 | 0.871 | | |
| **Difference CS – DE** | **–** | **−0.065** | **−0.037** | **0.051** | **0.075** | **0.020** | | |
| **Mean utility alive** | **DE patients** | **NL patients** | **ES patients** | **PT patients** | **FR patients** | **UK[1] patients** | **HU patients** | **RO patients** |
| *CONVINCE* | | | | | | | | |
| N | 223 | 56 | 315 | 137 | 126 | 36 | 110 | 357 |
| DE value set | 0.734 | 0.748 | 0.760 | 0.819 | 0.774 | 0.699 | 0.835 | 0.776 |
| CS value set[1] | 0.734 | 0.679 | 0.717 | 0.807 | 0.808 | 0.589 | 0.811 | 0.811 |
| **Difference CS – DE** | **–** | **−0.069** | **−0.043** | **−0.012** | **0.034** | **−0.110** | **−0.024** | **0.035** |

DE, Germany; NL, the Netherlands; ES, Spain; SE, Sweden; PL, Poland; AU, Australia; CS, country-specific; PT, Portugal; FR, France; UK, United Kingdom; HU, Hungary; RO, Romania.

[1]For the UK country-specific value set, the EQ-5D-3L crosswalk value is used.

For PREFERABLE-EFFECT, the difference between the highest and lowest incremental QALYs of the different value sets is 0.007 QALY (0.020–0.013 QALY: ES and SE value set). For CONVINCE, the difference between the highest and lowest incremental QALY in the trial-based analysis is 0.013 QALY (0.058–0.045: NL and RO value set). The CS value set resulted in a QALY gain of 0.017 (95%CI: −0.010–0.044) and 0.047 (95%CI: 0.004–0.090) for PREFERABLE and CON-VINCE respectively. The difference in incremental QALYs for the model-based analysis was 0.105 (0.980–0.875 QALY: FR and UK 3L value set).

Fig 3 shows the impact of value set on the probability of cost-effectiveness. For the trial-based analyses, the maximum difference in probability of cost-effectiveness is between the value sets with the maximum difference in incremental QALY (Table 3). For the CONVINCE model-based analysis, the maximum difference is between the value sets with the highest and lowest average utility score alive (Table 1). In PREFERABLE-EFFECT, the intervention was dominant, and the difference in probability between value sets is largest at higher willingness-to-pay (WTP) thresholds (e.g., 8.3% difference at €80.000/QALY). For CONVINCE, the incremental cost-effectiveness ratio (ICER) ranged between €41,121/QALY and €53,000/QALY for the trial-based analysis, and €37,471/QALY and €41,957 QALY for the model-based analysis. The difference in probability of cost-effectiveness is largest when the WTP threshold approximates the ICER ($\Delta_{max}$20.1% at WTP of €40.000/QALY for the model-based analysis).

## Scenario analyses

Complete results of all imputation scenarios are given in S6 & S7 Table in S1 File. The average of the difference between the highest and lowest utility alive gain using different value sets *across the imputation scenarios* was 0.013 (range: 0.011–0.016) for PREFERABLE-EFFECT. For CONVINCE, this was 0.008 (range: 0.007–0.010). The mean variation in utility alive gain across imputation scenarios *within one value set* was 0.004 for PREFERABLE-EFFECT (range: 0.002–0.006) and 0.003 for CONVINCE (range: 0.002–0.005).

## Discussion

Choice of EQ-5D-5L value set impacted the estimated absolute utility scores per year alive ($\Delta_{max}$0.127 and $\Delta_{max}$0.133 for PREFERABLE-EFFECT and CONVINCE, respectively). In the trial-based analyses, the included value sets resulted in a

**Table 3. Mean QALYs of the complete patient population per randomization group using different value sets.**

| | Mean QALY (i) | Mean QALY (c) | ΔE | (95%CI) | ΔC | (95%CI) | ICER |
|---|---|---|---|---|---|---|---|
| **PREFERABLE-EFFECT [1]** | | | | | | | |
| DE value set | 0.617 | 0.602 | 0.014 | (−0.014–0.041) | -€163 | (−€2,228 - €1,901) | Dominant |
| NL value set | 0.568 | 0.549 | 0.017 | (−0.011–0.044) | -€163 | (−€2,228 - €1,901) | Dominant |
| ES value set | 0.589 | 0.568 | 0.020 | (−0.007–0.046) | -€163 | (−€2,228 - €1,901) | Dominant |
| SE value set | 0.647 | 0.632 | 0.013 | (−0.014–0.040) | -€163 | (−€2,228 - €1,901) | Dominant |
| PL value set | 0.656 | 0.638 | 0.017 | (−0.010–0.043) | -€163 | (−€2,228 - €1,901) | Dominant |
| AU value set | 0.629 | 0.613 | 0.015 | (−0.013–0.042) | -€163 | (−€2,228 - €1,901) | Dominant |
| CS value set | 0.607 | 0.589 | 0.017 | (−0.010–0.044) | -€163 | (−€2,228 - €1,901) | Dominant |
| **Difference highest-lowest** | **0.088** | **0.089** | **0.007** | | | | |
| **CONVINCE – trial-based [2]** | | | | | | | |
| DE value set | 1.500 | 1.442 | 0.057 | (0.012–0.102) | €2,385 | (-€463 - €5,232) | €41,842 |
| NL value set | 1.377 | 1.319 | 0.058 | (0.014–0.102) | €2,385 | (-€463 - €5,232) | €41,121 |
| ES value set | 1.416 | 1.362 | 0.054 | (0.012–0.096) | €2,385 | (-€463 - €5,232) | €44,167 |
| PT value set | 1.478 | 1.428 | 0.050 | (0.007–0.094) | €2,385 | (-€463 - €5,232) | €47,700 |
| FR value set | 1.561 | 1.509 | 0.052 | (0.007–0.097) | €2,385 | (-€463 - €5,232) | €45,865 |
| UK 3L value set | 1.312 | 1.262 | 0.049 | (0.006–0.093) | €2,385 | (-€463 - €5,232) | €48,673 |
| UK 5L value set[3] | 1.446 | 1.409 | 0.058 | (0.012–0.093) | €2,385 | (-€463 - €5,232) | €45,000 |
| HU value set | 1.436 | 1.380 | 0.055 | (0.008–0.102) | €2,385 | (-€463 - €5,232) | €43,364 |
| RO value set | 1.559 | 1.514 | 0.045 | (0.003–0.086) | €2,385 | (-€463 - €5,232) | €53,000 |
| CS value set | 1.482 | 1.435 | 0.047 | (0.004–0.090) | €2,385 | (-€463 - €5,232) | €50,745 |
| **Difference highest-lowest** | **0.249** | **0.252** | **0.013** | | | | **€11,879** |
| **CONVINCE – model-based** | | | | | | | |
| DE value set | 9.714 | 8.738 | 0.976 | (0.077–1.854) | €36,730 | (€2,370; €73,861) | €37,627 |
| NL value set | 9.206 | 8.276 | 0.931 | (0.101–1.772) | €36,730 | (€2,370; €73,861) | €39,463 |
| FR value set | 9.970 | 8.990 | 0.980 | (0.070–1.898) | €36,730 | (€2,370; €73,861) | €37,471 |
| UK 3L value set | 8.936 | 8.061 | 0.875 | (0.098–1.681) | €36,730 | (€2,370; €73,861) | €41,957 |
| RO value set | 9.961 | 9.009 | 0.952 | (0.035–1.859) | €36,730 | (€2,370; €73,861) | €38,576 |
| CS value set | 9.643 | 8.711 | 0.932 | (0.078–1.812) | €36,730 | (€2,370; €73,861) | €39,429 |
| **Difference highest-lowest** | **1.034** | **0.948** | **0.105** | | | | **€4,487** |

QALY, quality-adjusted life year; (i), intervention group; (c), control group; ΔE, difference in effects (QALY intervention – QALY control); CI, Confidence Interval; ICER, Incremental Cost-Effectiveness Ratio; WTP, willingness-to-pay; DE, Germany; NL, the Netherlands; ES, Spain; SE, Sweden; PL, Poland; AU, Australia; CS, country-specific; PT, Portugal; FR, France; UK, United Kingdom; HU, Hungary; RO, Romania.

[1]Control variables PREFERABLE-EFFECT: baseline utility for QALY estimates; treatment center and line of treatment for both QALY and cost estimates.

[2]Control variables CONVINCE: baseline utility for QALY estimates, baseline costs for cost estimates, country for both QALY and cost estimates.

[3]For the UK, the crosswalk version of the EQ-5D-3L value set was applied using the index values at the EuroQol website, following the national guide-line. As an additional analysis, we also applied the 2026 EQ-5D-5L UK value set, although this is not officially adopted for national use at time of writing. For the country-specific value set, the EQ-5D-3L crosswalk value is used.

maximum difference in the probability of cost-effectiveness of 8.3% at a WTP of €80,000/QALY in PREFERABLE-EFFECT and 11.1% at €40,000/QALY for CONVINCE. In the model-based analysis of CONVINCE, the maximum difference in incremental QALYs was 0.105, and the maximum difference in probability of cost-effectiveness was 20.1% at a WTP of €40,000/QALY. These results suggest that cost-utility outcomes may be impacted by the choice of value set in different patient populations.

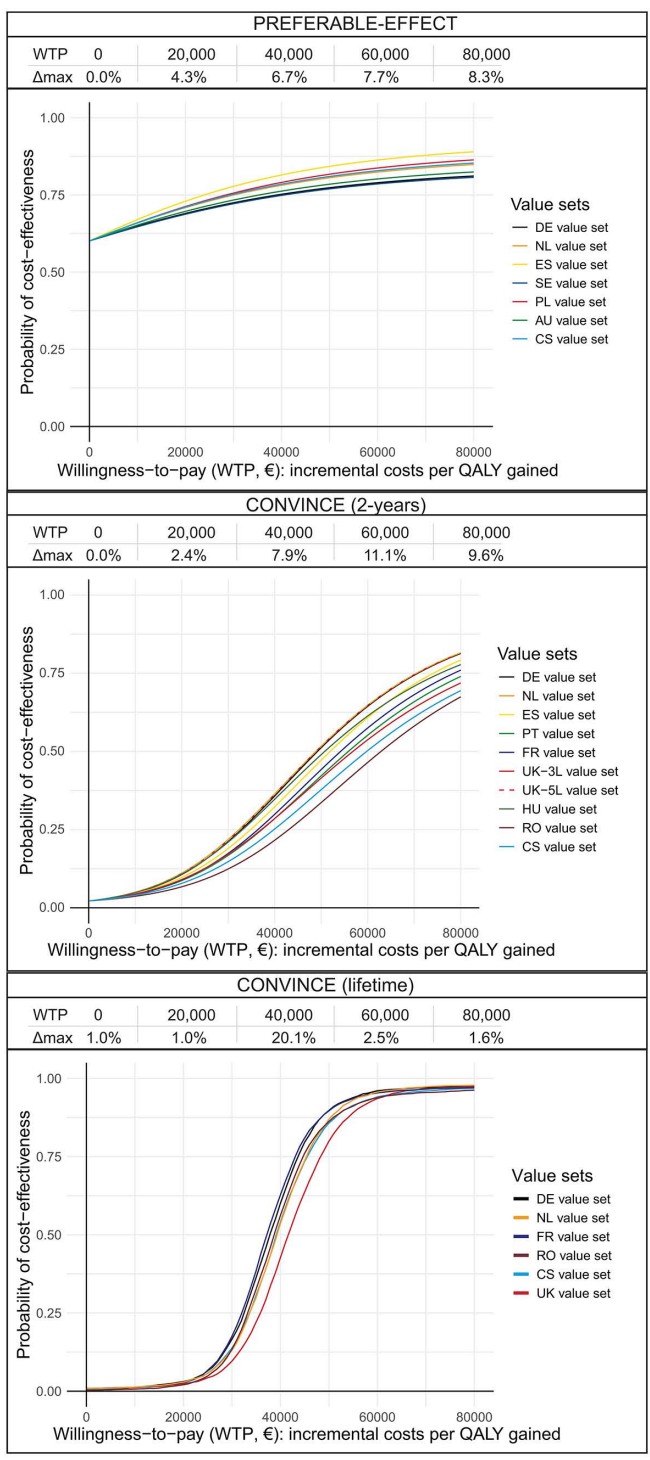

**Fig 3. Cost-effectiveness acceptability curves and probability of cost-effectiveness.** For the United Kingdom, the crosswalk version of the EQ-5D-3L value set was applied using the index values at the EuroQol website, following the national guideline. As an additional analysis, we also applied the 2026 EQ-5D-5L United Kingdom value set, although this is not officially adopted for national use at time of writing. For the country-specific value set, the EQ-5D-3L crosswalk value is used. QALY, quality-adjusted life year; DE, Germany; NL, the Netherlands; ES, Spain; SE, Sweden; PL, Poland; AU, Australia; CS, country-specific; PT, Portugal; FR, France; UK, United Kingdom; HU, Hungary; RO, Romania.

Consistent with previous research, our results showed substantial variation in average utility scores between the value sets (maximum difference in utility score per year alive of 0.127 and 0.133 for PREFERABLE and CONVINCE, respectively) [16,21–31], with largest variation at the lower utility scores [16,22,25,27,28,61,62]. Similar to the findings of Oppong et al. (2011) and van Dongen et al. (2021), we found that for the trial-based analyses, value sets that resulted in higher absolute utility scores did not directly lead to a higher or lower incremental difference between intervention and control [16,23]. The variation in QALY gains is primarily driven by the EQ-5D-5L domains impacted and related country-specific preferences [26]. In the lifetime Markov cohort model of the CONVINCE trial, however, higher utility scores did result in larger QALY losses for the same reduction in life-years. This correlation indicates that choice of value set is more impactful when mortality differences are large. These findings reinforce the ISPOR Task Force recommendation that when several utility scores are used in an economic model, they should as much as possible be elicited using the same methods, population, and value set [63].

The UK crosswalk value set has been suggested for multinational analyses and when a national value set is missing [6,16]. This value set resulted in much lower average utility scores compared to those of other countries [23,25,27–30], as well as a lower probability of cost-effectiveness in the model-based analysis. The maximum difference in probability was 20.1% at a WTP threshold of €40,000, compared to 9.1% when this outlier was excluded. This difference in probability implies that applying the UK crosswalk value set might underestimate the probability of cost-effectiveness [22]. Although the 2026 UK EQ-5D-5L value set is not officially adopted for national use at time of writing [56], the additional analysis showed that estimates based on this set did fall between those of the other value sets. The EQ-5D-3L value set is likely obsolete, as it was developed using methods currently considered outdated, and the underlying population preferences may have changed since 1990 [64,65]. Therefore, we recommend against using the UK EQ-5D-3L crosswalk value set to estimate pooled cost-utility outcomes alongside a multinational trial.

General cost-effectiveness principles apply. When interventions are expected to be both cost-saving and effective, the impact of choice of value set will be considerably less than when the intervention yields higher QALYs & higher costs, especially when the ICER is close to the willingness-to-pay threshold [16,29]. For example, if the willingness-to-pay threshold was €50,000/QALY gained, then different value sets could have resulted in different cost-effectiveness conclusions for the CONVINCE trial-based analyses. The probability of cost-effectiveness will be more strongly impacted by choice of value set when the uncertainty surrounding QALYs is smaller, than when there is a lot of uncertainty. Furthermore, the sensitivity analysis showed that different imputation strategies resulted in a change of utility gain per year alive. This finding highlights that although the impact of choice of value set is relevant, other factors also influence the exact value of the estimated treatment effect, such as imputation strategy when there is a substantial number of missing values. Acknowledging these caveats improves interpretation and must be considered when comparing QALY outcomes between value sets.

## Strengths and limitations

This study has several strengths and limitations to consider when interpreting the findings. One strength is the use of two empirical multinational studies, thereby evaluating the impact of different interventions, patient populations, and time horizons. One limitation of this study is that the value sets of the countries included in these trials follow similar preference patterns [7]. The impact of choice of value set might be more substantial in multinational trials with countries whose preferences deviate more, as shown by van Dongen et al (2021) [23]. As a result, these findings might not hold outside the European context. Another consideration is that this study implicitly assumes that there is measurement equivalence, i.e., the answers to the EQ-5D-5L questionnaire are interpreted the same across countries, and are not influenced by, e.g., cultural differences in self-reporting. Whether this assumption is valid is up for debate [24,66–68], but falls outside of the scope of this article. Finally, the number of included participants per country differs for both studies. The responses from countries with higher recruitment numbers have a larger influence on the estimated pooled values than countries with lower participation.

## Implications

While our findings are similar to Oppong et al. (2011) [16], we do not fully agree with their conclusion that choice of value set does not matter for cost-utility outcomes. Most variation between value sets occurs at the lower utility scores. Consequently, the impact of choice of value set might be stronger for interventions preventing or aiding recovery of health conditions associated with low quality of life, e.g., prevention of moderate to severe strokes [69], than for supportive interventions for patients with chronic diseases, such as PREFERABLE-EFFECT and CONVINCE. This hypothesis is supported by the study conducted by Karlsson et al. on rheumatoid arthritis patients [25], where treatment improved the low baseline utility scores substantially. Karlsson et al. found that the 1-year adjusted QALY gain ranged between 0.06 and 0.09 when using the US and UK value set respectively. Future research should further investigate which underlying factors and conditions are determinants for changes in outcomes, e.g., by means of a simulation study.

The maximum difference in probability of cost-effectiveness was around 10% for both trial-based analyses, as well as for the model-based analysis when excluding the outlier value. For these empirical studies, using country-specific value sets, or selecting a value set that results in QALY gains closest to the average across all sets, appears to be a pragmatic and appropriate approach. However, if the aim is to inform national reimbursement decisions [15], applying country-specific preferences is essential to allow for comparability across interventions [70]. Health economic evaluations will likely continue to require localization, as these results are generally not considered transferable between countries and are therefore excluded from the European Union's joint clinical assessment framework of health technologies [71]. For future multinational health economic analyses, we recommend performing sensitivity analyses using multiple value sets to see the impact on outcomes. As discussed above, previous research suggests that when (1) preferences patterns between included countries are dissimilar, (2) the intervention substantially improves or prevents severe health states, and/or (3) mortality is strongly impacted by the intervention, the choice of value set may have a stronger influence on outcomes. Being aware of the potential impact is crucial, as over- or underestimating quality-of-life improvements due to the choice of value set may lead to suboptimal policy or clinical decisions.

## Conclusion

The choice of EQ-5D-5L value set led to substantial variation in absolute utility scores and QALYs. In these two multinational studies of supportive interventions for patients with chronic diseases, the maximum difference in probability of cost-effectiveness was around 10%. For short-term analyses, applying country-specific value sets or selecting a single value set that yields QALY estimates close to the average across all options appears to be a pragmatic and appropriate choice. These findings suggest that the commonly used UK value set may be less appropriate, as it consistently resulted in lower utility scores compared to the other value sets. The impact on cost-utility outcomes may be larger in interventions resulting in large mortality differences, if the intervention prevents or aids recovery of health conditions associated with low quality of life, and/or when the preferences patterns between included countries are very dissimilar. Scenario analyses using multiple value sets should be incorporated into a checklist for multinational trials, as over- or underestimating quality-of-life improvements may lead to suboptimal policy or clinical decisions.

## Supporting information

**S1 File.** Supplementary Material EQ5D PLOS One. Contains Supplementary Table S1-S7 and Supplementary Figure S1-S2.
(DOCX)

## Acknowledgments

The authors thank all participants and involved professional staff at all participating centers for their contribution to the PREFERABLE-EFFECT and CONVINCE randomized controlled trials.

PREFERABLE-EFFECT Scientific Committee:

• Anne May (lead author), Department of Epidemiology and Health Economics, Julius Center for Health Sciences and Primary Care, University Medical Center Utrecht, Utrecht University, Utrecht, The Netherlands, a.m.may@umcutrecht.nl

• Ander Urroticoechearibate, Gipuzkoa Cancer Unit, OSID-Onkologikoa, BioGipuzkoa, Osakidetza, San Sebastian, Spain

• Eva Zopf, Mary MacKillop Institute for Health Research, Australian Catholic University, Melbourne, Australia AND Cabrini Cancer Institute, Cabrini Health, Melbourne, Victoria, Australia

• Joachim Wiskemann, Working Group Exercise Oncology, Division of Medical Oncology, Heidelberg University Hospital and NCT Heidelberg, a partnership between DKFZ and University Medical Center Heidelberg, Heidelberg, Germany

• Karen Steindorf, Division of Physical Activity, Prevention and Cancer, German Cancer Research Center (DKFZ) and National Center for Tumor Diseases (NCT) Heidelberg, a partnership between DKFZ and University Medical Center Heidelberg, Heidelberg, Germany

• Martijn Stuiver, Division of Psychosocial Research and Epidemiology & Center for Quality of Life, Netherlands Cancer Institute, Amsterdam, The Netherlands

• Wilhelm Bloch, Department of Molecular and Cellular Sports Medicine, Institute of Cardiovascular Research and Sports Medicine, German Sport University Cologne, Cologne, Germany

• Yvonne Wengstrom, Karolinska Institutet and Karolinska Comprehensive Cancer Center, Karolinska University Hospital, Stockholm, Sweden

• Elzbieta Senkus, Department of Oncology and Radiotherapy, Medical University of Gdańsk, Gdańsk, Poland

CONVINCE Scientific Committee:

• Michiel Bots (lead author), Department of Epidemiology and Health Economics, Julius Center for Health Sciences and Primary Care, University Medical Center Utrecht, Utrecht University, Utrecht, The Netherlands, M.L.Bots@umcutrecht.nl

• Claudia Barth, Medical Scientific Affairs, B. Braun Avitum, Melsungen, Germany

• Peter Blankestijn, Department of Nephrology and Hypertension, University Medical Center Utrecht, Utrecht University, Utrecht, the Netherlands

• Bernard Canaud, Fresenius Medical Care Deutschland GmbH, Global Medical Office, Bad Homburg, Germany; Montpellier University School of Medicine, Montpellier, France

• Krister Cromm, Fresenius Medical Care Deutschland GmbH, Global Medical Office, Bad Homburg, Germany

• Andrew Davenport, Department of Renal Medicine, Division of Medicine, Royal Free Hospital London, University College London, London, England

• Kathrin Fischer, Centre of Internal Medicine and Dermatology, Department of Psychosomatic Medicine, Charité Universitätsmedizin Berlin, corporate member of Freie Universität Berlin, Humboldt-Universität zu Berlin, and Berlin Institute of Health, Berlin, Germany

• Jörgen Hegbrant, Division of Nephrology, Department of Clinical Sciences, Lund University, Lund, Sweden

• Matthias Rose, Center for Patient-Centered Outcomes Research and Centre of Internal Medicine and Dermatology, Department for Psychosomatic Medicine, Charité – Universitätsmedizin Berlin, corporate member of Freie Universität Berlin and Humboldt Universität zu Berlin, Berlin, Germany

• Giovanni Strippoli, The Department of Precision and Regenerative Medicine and Ionian Area, University of Bari, Bari, Italy; The School of Public Health, University of Sydney, Sydney, Australia

• Mariëtta Török, Corporate Medical Office Diaverum, Malmö, Sweden

• Mark Woodward, George Institute for Global Health, School of Public Health, Imperial College London, London, England; George Institute for Global Health, University of New South Wales, Sydney, Australia

## Author contributions

**Conceptualization:** Aniek E.M. Schouten, Miriam P. van der Meulen.

**Formal analysis:** Aniek E.M. Schouten.

**Methodology:** Aniek E.M. Schouten, Miriam P. van der Meulen.

**Supervision:** Geert W.J. Frederix, Anne M. May, Miriam P. van der Meulen.

**Writing – original draft:** Aniek E.M. Schouten, Miriam P. van der Meulen.

**Writing – review & editing:** Aniek E.M. Schouten, Geert W.J. Frederix, Martijn M. Stuiver, Felix Fischer, Anouk E. Hiensch, Krister Cromm, Bernard Canaud, Giovanni F.M. Strippoli, Anne M. May, Miriam P. van der Meulen.

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
