## [Decision Letter · Decision Letter 0]

1 Apr 2026

PONE-D-25-69177Choice and Impact of EQ-5D-5L Value Set in Cost-Utility Analyses Alongside Multinational Trials: Insights From PREFERABLE-EFFECT and CONVINCEPLOS One

Dear Dr. Schouten,

Thank you for submitting your manuscript to PLOS ONE. After careful consideration, we feel that it has merit but does not fully meet PLOS ONE’s publication criteria as it currently stands. Therefore, we invite you to submit a revised version of the manuscript that addresses the points raised during the review process.

We look forward to receiving your revised manuscript.

Kind regards,

Henry Hugh Bailey, Ph.D

Academic Editor

PLOS One

**Journal Requirements:**

1. When submitting your revision, we need you to address these additional requirements. Please ensure that your manuscript meets PLOS ONE's style requirements, including those for file naming. The PLOS ONE style templates can be found at https://journals.plos.org/plosone/s/file?id=wjVg/PLOSOne_formatting_sample_main_body.pdf and https://journals.plos.org/plosone/s/file?id=ba62/PLOSOne_formatting_sample_title_authors_affiliations.pdf 2. Thank you for stating the following in the Acknowledgments Section of your manuscript: PREFERABLE-EFFECT was supported by the European Union’s Horizon 2020 research and innovation program (grant agreement 825677) and the National Health and Medical Research Council of Australia (2018/GNT1170698). CONVINCE was exclusively supported by a grant from the European Commission (Horizon 2020 program, grant agreement 754803). The authors thank all participants and involved professional staff at all participating centers for their contribution to the PREFERABLE-EFFECT and CONVINCE randomized controlled trials. We note that you have provided funding information that is not currently declared in your Funding Statement. However, funding information should not appear in the Acknowledgments section or other areas of your manuscript. We will only publish funding information present in the Funding Statement section of the online submission form. Please remove any funding-related text from the manuscript and let us know how you would like to update your Funding Statement. Currently, your Funding Statement reads as follows: The author(s) received no specific funding for this work. Please include your amended statements within your cover letter; we will change the online submission form on your behalf. 3. Thank you for stating the following in the Competing Interests section: I have read the journal's policy and the authors of this manuscript have the following competing interests: GF reports a consulting role for Illumina (Inst). KC reports holding Fresenius Medical Care company stocks (ordinary shares) and being an employee of Fresenius Medical Care. BC reports being CEO of MTX Consulting Int, Montpellier; Emeritus Medical Officer at FMC Deutschland Global Medical Office; and Emeritus Professor of Medicine, Montpellier, France. GS reports receiving honoraria for educational events from Fresenius Medical Care Deutschland GmbH. All other authors declared no competing interests.  We note that one or more of the authors are employed by a commercial company.  a. Please provide an amended Funding Statement declaring this commercial affiliation, as well as a statement regarding the Role of Funders in your study. If the funding organization did not play a role in the study design, data collection and analysis, decision to publish, or preparation of the manuscript and only provided financial support in the form of authors' salaries and/or research materials, please review your statements relating to the author contributions, and ensure you have specifically and accurately indicated the role(s) that these authors had in your study. You can update author roles in the Author Contributions section of the online submission form. Please also include the following statement within your amended Funding Statement. “The funder provided support in the form of salaries for authors, but did not have any additional role in the study design, data collection and analysis, decision to publish, or preparation of the manuscript. The specific roles of these authors are articulated in the ‘author contributions’ section.”If your commercial affiliation did play a role in your study, please state and explain this role within your updated Funding Statement.  b. Please also provide an updated Competing Interests Statement declaring this commercial affiliation along with any other relevant declarations relating to employment, consultancy, patents, products in development, or marketed products, etc.   Within your Competing Interests Statement, please confirm that this commercial affiliation does not alter your adherence to all PLOS ONE policies on sharing data and materials by including the following statement: "This does not alter our adherence to  PLOS ONE policies on sharing data and materials.” (as detailed online in our guide for authors http://journals.plos.org/plosone/s/competing-interests) . If this adherence statement is not accurate and  there are restrictions on sharing of data and/or materials, please state these. Please note that we cannot proceed with consideration of your article until this information has been declared. Please include both an updated Funding Statement and Competing Interests Statement in your cover letter. We will change the online submission form on your behalf. 4. Thank you for stating the following in the Competing Interests section: I have read the journal's policy and the authors of this manuscript have the following competing interests: GF reports a consulting role for Illumina (Inst). KC reports holding Fresenius Medical Care company stocks (ordinary shares) and being an employee of Fresenius Medical Care. BC reports being CEO of MTX Consulting Int, Montpellier; Emeritus Medical Officer at FMC Deutschland Global Medical Office; and Emeritus Professor of Medicine, Montpellier, France. GS reports receiving honoraria for educational events from Fresenius Medical Care Deutschland GmbH. All other authors declared no competing interests.  Please confirm that this does not alter your adherence to all PLOS ONE policies on sharing data and materials, by including the following statement: "This does not alter our adherence to  PLOS ONE policies on sharing data and materials.” (as detailed online in our guide for authors http://journals.plos.org/plosone/s/competing-interests). If there are restrictions on sharing of data and/or materials, please state these. Please note that we cannot proceed with consideration of your article until this information has been declared.  Please include your updated Competing Interests statement in your cover letter; we will change the online submission form on your behalf. 5. We note that you have indicated that there are restrictions to data sharing for this study. For studies involving human research participant data or other sensitive data, we encourage authors to share de-identified or anonymized data. However, when data cannot be publicly shared for ethical reasons, we allow authors to make their data sets available upon request. For information on unacceptable data access restrictions, please see http://journals.plos.org/plosone/s/data-availability#loc-unacceptable-data-access-restrictions.  Before we proceed with your manuscript, please address the following prompts: a) If there are ethical or legal restrictions on sharing a de-identified data set, please explain them in detail (e.g., data contain potentially identifying or sensitive patient information, data are owned by a third-party organization, etc.) and who has imposed them (e.g., a Research Ethics Committee or Institutional Review Board, etc.). Please also provide contact information for a data access committee, ethics committee, or other institutional body to which data requests may be sent. b) If there are no restrictions, please upload the minimal anonymized data set necessary to replicate your study findings to a stable, public repository and provide us with the relevant URLs, DOIs, or accession numbers. Please see http://www.bmj.com/content/340/bmj.c181.long for guidelines on how to de-identify and prepare clinical data for publication. For a list of recommended repositories, please see https://journals.plos.org/plosone/s/recommended-repositories. You also have the option of uploading the data as Supporting Information files, but we would recommend depositing data directly to a data repository if possible. Please update your Data Availability statement in the submission form accordingly. 6. One of the noted authors is a group or consortium [the PREFERABLE-EFFECT Scientific Committee, and the CONVINCE Scientific Committee]. In addition to naming the author group, please list the individual authors and affiliations within this group in the acknowledgments section of your manuscript. Please also indicate clearly a lead author for this group along with a contact email address. 7. We note that there is identifying data in the Supporting Information file <Supplementary-Material_EQ5D_PLOS-One.docx>. Due to the inclusion of these potentially identifying data, we have removed this file from your file inventory. Prior to sharing human research participant data, authors should consult with an ethics committee to ensure data are shared in accordance with participant consent and all applicable local laws. Data sharing should never compromise participant privacy. It is therefore not appropriate to publicly share personally identifiable data on human research participants. The following are examples of data that should not be shared: -Name, initials, physical address-Ages more specific than whole numbers-Internet protocol (IP) address-Specific dates (birth dates, death dates, examination dates, etc.)-Contact information such as phone number or email address-Location data-ID numbers that seem specific (long numbers, include initials, titled “Hospital ID”) rather than random (small numbers in numerical order) Data that are not directly identifying may also be inappropriate to share, as in combination they can become identifying. For example, data collected from a small group of participants, vulnerable populations, or private groups should not be shared if they involve indirect identifiers (such as sex, ethnicity, location, etc.) that may risk the identification of study participants. Additional guidance on preparing raw data for publication can be found in our Data Policy (https://journals.plos.org/plosone/s/data-availability#loc-human-research-participant-data-and-other-sensitive-data) and in the following article: http://www.bmj.com/content/340/bmj.c181.long. Please remove or anonymize all personal information (<specific identifying information in file to be removed>), ensure that the data shared are in accordance with participant consent, and re-upload a fully anonymized data set. Please note that spreadsheet columns with personal information must be removed and not hidden as all hidden columns will appear in the published file. 8. Please include captions for your Supporting Information files at the end of your manuscript, and update any in-text citations to match accordingly. Please see our Supporting Information guidelines for more information: http://journals.plos.org/plosone/s/supporting-information. 9. If the reviewer comments include a recommendation to cite specific previously published works, please review and evaluate these publications to determine whether they are relevant and should be cited. There is no requirement to cite these works unless the editor has indicated otherwise.

Reviewers' comments:

Reviewer's Responses to Questions

**Comments to the Author**

1. Is the manuscript technically sound, and do the data support the conclusions?

Reviewer #1: Yes

Reviewer #2: Yes

2. Has the statistical analysis been performed appropriately and rigorously? 

Reviewer #1: Yes

Reviewer #2: Yes

3. Have the authors made all data underlying the findings in their manuscript fully available?

Reviewer #1: Yes

Reviewer #2: Yes

4. Is the manuscript presented in an intelligible fashion and written in standard English?

Reviewer #1: Yes

Reviewer #2: Yes

5. Review Comments to the Author

**Reviewer #1:** Dear author:

I greatly enjoyed reading this manuscript and learned a lot from your analysis. The topic is highly relevant and the methodological approach is carefully conducted. I only have several minor comments that may help improve clarity and readability for a broader audience.

Major comments:

1. I suggest that the research gap be articulated more explicitly in the objective section. While the study aim is clearly stated, it would strengthen the abstract to briefly clarify what remains uncertain in the existing literature and how this study addresses that gap.

2. The two trials (PREFERABLE-EFFECT and CONVINCE) involve different patient populations and disease contexts. It would be helpful to briefly clarify whether differences in demographics or baseline disease severity between these trials may influence the observed variation in utility estimates across value sets.

3. The manuscript states: “Health economic evaluation guidelines recommend using the value set of the jurisdiction of interest. However, there is currently no guideline on which value set(s) should be used…” This statement could benefit from clarification. Since national guidelines do provide recommendations for jurisdiction-specific analyses, it would be helpful to more clearly explain the gap that exists specifically in the context of pooled multinational analyses.

4. The statement that “there is no standard practice for estimating utilities” could be further elaborated. I suggest that the authors briefly describe how pooled multinational analyses are currently conducted in practice to provide clearer context.

5. The manuscript mentions that the study includes patients with different chronic conditions. It may be helpful to clarify whether differences in disease severity or symptom burden were considered as potential modifiers of the impact of value set choice.

6. The sentence “Utility scores of the full sample were calculated with the value sets of each included country, and with the country-specific value set of each participant” is somewhat unclear. Could the authors please clarify exactly how these two approaches differ in practice? Specifically, does “value sets of each included country” mean that the entire pooled sample was re-valued multiple times using each country's tariff separately?

7. It would be helpful if the authors could further justify the selection of the PREFERABLE-EFFECT and CONVINCE trials. As the two studies involve distinct patient populations (one predominantly female, one predominantly male), it may be useful to clarify the rationales.

8. “The imputed datasets were analyzed using Rubin’s rules” could you please provide additional explanations on Rubin’s rule?

9. I am slightly confused as to whether the authors applied both the crosswalk version of the EQ-5D-3L and the EQ-5D-5L UK value set in the main analyses.

10. In Table 2, the number of respondents differs substantially across countries, and the variation appears quite large. Have the authors examined whether these differences in sample size may influence the estimated utility values?

Minor comments

1 in Line 126, the abbreviations of ‘Netherlands, Germany, Spain,’ are lacked.

**Reviewer #2:** Thank you for the opportunity to review this manuscript.

This study investigates how the choice of EQ-5D-5L value sets affect cost-utility outcomes using two multinational trials. The authors compare utility scores and QALY estimated using a single value set against those derived from country specific value sets. Below, I provide some comments that I believe will improve the quality of the manuscript.

- Line 83-84: The introduction seems to have a contradicting problem statement. Specifically, the following two sentences “Health economic evaluation guidelines recommend using the value set of the jurisdiction of interest. However, there is currently no guideline on which value set(s) should be used for these types of analyses to estimate utility scores.” I think it might be helpful for authors to clarify whether the gap lies in the absence of guidelines for multinational pooled analysis, or a broader absence of consensus. I believe clarifying this point is important as it is the core rationale for the study.

- Line 95: The authors mention that “Oppong et al. (2010) found that the choice between European, country-specific, or UK value sets in country-specific sub-group analyses had little impact on cost-utility outcomes when using the EQ-5D-3L.” I think authors should discuss how the findings of this study relate to the findings of Oppong et al.

- To strengthen its policy relevance, I would suggest adding the European policy context and broader European HTA landscape. The recent EU HTA regulation has a direct implication for how pooled data should be handled.

- Missing data in CONVINCE is concerningly high. I wonder if the authors have some possible explanations for that.

- Lines 144-147, it is quite difficult to follow what is being imputed/calculated, I assume mainly due to the possibility of different terminologies being used across fields. It would be helpful if the authors consider using some mathematical formulas for clarity, which can then be put in the appendix.

- If the authors would like to keep the paper up to date, the UK value set for the EQ-5D-5L is now available (https://doi.org/10.1016/j.jval.2026.03.008 )

6. PLOS authors have the option to publish the peer review history of their article (what does this mean?). If published, this will include your full peer review and any attached files.

Reviewer #1: No

Reviewer #2: No

---

## [Author Response · Author response to Decision Letter 1]

5 May 2026

May 4, 2026

Manuscript ID: PONE-D-25-69177

Manuscript title: Choice and Impact of EQ-5D-5L Value Set in Cost-Utility Analyses Alongside Multinational Trials: Insights From PREFERABLE-EFFECT and CONVINCE

Thank you for considering our paper for publication in PLOS One, and the opportunity to address the comments from the reviewers. We incorporated most of their feedback and agree that these improve the quality of our manuscript. The changes in the manuscript are highlighted with ‘track changes’. Please find our detailed responses on all comments in the response to reviewers document. In the cover letter, changes to the Funding Statement, Acknowledgements Section and Competing Interests Statement based on the journal requirements can be found.

Kind regards,

Aniek Schouten

---

## [Editor Report · Decision Letter 1]

7 May 2026

Choice and Impact of EQ-5D-5L Value Set in Cost-Utility Analyses Alongside Multinational Trials: Insights From PREFERABLE-EFFECT and CONVINCE

PONE-D-25-69177R1

Dear Dr. Schouten,

We are pleased to inform you that your manuscript has been judged scientifically suitable for publication and will be formally accepted for publication once it meets all outstanding technical requirements.

Kind regards,

Henry Bailey, Ph.D

Academic Editor

PLOS One
---

## [Editor Report · Acceptance letter]

PONE-D-25-69177R1

PLOS One

Dear Dr. Schouten,

I'm pleased to inform you that your manuscript has been deemed suitable for publication in PLOS One. Congratulations! Your manuscript is now being handed over to our production team.

Kind regards,

on behalf of

Dr. Henry Hugh Bailey

Academic Editor

PLOS One